# Fabrication and Characterization of Waste Wood Cellulose Fiber/Graphene Nanoplatelet Carbon Papers for Application as Electromagnetic Interference Shielding Materials

**DOI:** 10.3390/nano11112878

**Published:** 2021-10-28

**Authors:** Jihyun Park, Lee Ku Kwac, Hong Gun Kim, Hye Kyoung Shin

**Affiliations:** Institute of Carbon Technology, Jeonju University, 303 Cheonjam-ro, Wansan-gu, Jeonju-si 55069, Jeollabuk-do, Korea; jennai@jj.ac.kr (J.P.); kwack29@jj.ac.kr (L.K.K.)

**Keywords:** waste wood, electromagnetic interference shielding, graphene nanoplatelet, carbon papers

## Abstract

Waste wood contains large amounts of cellulose fibers that have outstanding mechanical properties. These fibers can be recycled and converted into highly valuable materials of waste wood. In this study, waste wood cellulose fiber/graphene nanoplatelet (WWCF/GnP) papers were prepared according to the WWCF and GnP contents. Subsequently, the WWCF/GnP papers were varyingly carbonized for their application as electromagnetic interference (EMI) shielding materials such as state-of-the-art electronic equipment malfunction prevention, chip-level microsystem, and micro intersystem noise suppression/reduction. The increase in the GnP content and carbonization temperature enhanced electrical conductivity, thereby generating a greater EMI shielding effectiveness (EMI SE) in the high-frequency X-band. Additionally, the thickness of the WWCF/GnP carbon papers improved the electrical conductivity and EMI SE values. The electrical conductivity of the WWCF/GnP-15 carbon paper obtained at carbonization temperature of 1300 °C was approximately 5.86 S/m, leading to an EMI SE value of 43 decibels (dB) at 10.5 GHz for one sheet. Furthermore, overlapping of the three sheets increased the electrical conductivity to 7.02 S/m, leading to an EMI SE value of 72.5 dB at 10.5 GHz. Thus, we isolated WWCFs, without completely removing contaminants, for recycling and converting them into highly valuable EMI shielding materials.

## 1. Introduction

Consumption of fossil fuels and generation of wastes has led to severe environmental problems worldwide. Various types of waste wood used in industrial, construction, or household application are mostly disposed in landfills or incinerated in mixed waste forms. Waste wood consists of contaminants such as paint, oil, and glue that are added during wood processing, leading to environmental pollution [1,2,3,4,5,6,7,8]. However, waste wood also contains large amounts of cellulose fibers (CF). These CF of semi-crystalline fibers have outstanding mechanical properties owing to the high aspect ratios of fibers and have been utilized in papers, composites, and coatings [9,10,11]. Thus, highly valuable materials of waste wood can be obtained, irrespective of complete removal of contaminants, by recycling and converting the CF. These materials can be further researched as electromagnetic interference (EMI) shielding materials.

EMI shielding materials are known to prevent the shortening of the electronic machine lifetime, thereby protecting human health from exposure to electromagnetic (EM) radiation such as serious EM and heat emissions [12,13,14,15]. However, effective EMI shielding materials require high electrical conductivity, corrosion resistivity, absorption tunability, cost-effectivity, and flexibility [16,17,18,19]. Although metals, as representative conductive fillers, are widely consumed for EMI shielding applications [20,21,22,23], their native rigidity, easy corrosion, and insufficient dispersion have limits [24,25]. Hence, it is essential to replace these metals with other efficient materials. Graphene nanoplatelets (GnPs) are carbon materials composed of small stacks of graphene sheets with significant thickness ranging from one to tens of nanometers. These GnPs are lightweight and have high electrical conductivity, low density, and corrosion resistivity. Additionally, GnPs have a high specific surface area, enhancing the EM wave absorption due to multiple reflections. Moreover, these materials have a low cost as compared to metals. Thus, GnPs are considered potential candidates for effective EM defense [26,27,28,29,30]. Currently, the merits of GnPs have been gradually researched as EMI shielding materials. For example, Kashi et al. [31] prepared poly lactide/GnP nanocomposites for EMI shielding performance. As a result, the EMI shielding values of the nanocomposites were effective at the C- and X-bands. Furthermore, the addition of GnP improved the electrical permittivity and conductivity in poly lactide, resulting in a higher shielding effectiveness. Bhosale et al. [32] fabricated poly(ether ketone) (PEK)—GnP nanocomosites investigated their EMI shielding effectiveness (EMI SE) in the X-band. The resultant PEK-5 vol % nanocomoposites having 1 mm thickness showed an EMI SE of ≈33 decibels at an electrical conductivity of approximately 0.02 S/cm. Aïssa et al. [33] demonstrated the potential of a two-dimensional Ti_3_C_2_T*_x_* MXene/GnP composite with a sandwich-like structure and minimal thickness for EMI SE in the extreme high-frequency M-band ranging from 60 to 80 GHz. Although the supplement of 2.5 wt % GnPs decreased the composite surface roughness, the electrical conductivity (≈10^5^ S/cm) and Hall carrier mobility (55 cm^2^/V) increased. Therefore, the Ti_3_C_2_T_x_ MXene/GnPs film that was 1.75 μm thick represented an excellent absorbance of 64 dB. Liu et al. [34] prepared Co-doped Ni-Zn ferrite (CNZF)/graphene nanocomposites (GN) by hydrothermal method. In this study, GN acted to significantly affect the impedance matching and attenuation potential of the absorbers. Therefore, CNZF/GN nanocomposite can be a new type of microwave absorber with light weight.

In this study, waste wood CFs (WWCFs) were isolated, without completely removing the contaminants, and recycled and converted into EMI shielding materials. We isolated WWCFs via alkali cooking and bleaching treatment, and fabricated WWCF/GnP papers according to the WWCF and GnP contents. Moreover, the EMI SE of the WWCFs/GnPs paper was enhanced by carbonization at various temperatures because cellulose is a nonconductive material. The obtained carbon papers were identified by X-ray diffraction (XRD), Raman spectral analysis, and scanning electron microscopy (SEM). Furthermore, the tensile strength of carbon papers was determined. Additionally, the EMI shielding efficiency was recorded in the ranging from 8.2 to 12.4 GHz of X-band frequency.

## 2. Materials and Methods

### 2.1. Materials

Waste wood (plywood type) for this study was obtained from a waste wood disposal plant at Jeonju University (Jeonju-si, Korea). GnPs were purchased from Nanografi Nano Technology (Çankaya, Turkey). All chemicals were of analytical grade.

### 2.2. Preparation of WWCF/GnP Papers and Their Carbon Papers

Waste wood was chipped to approximately 10 × 10 cm and alkali-cooked using 15 wt % NaOH solution for 5 h at 121 °C in an autoclave. The alkali-cooked waste wood chips were beaten by a pestle to separate them into fibers, and then they were bleached using 10 wt % H_2_O_2_ solution and 5 wt % H_2_O_2_ stabilizer for 1 h at 80 °C to obtain CFs. The obtained bleached pulp was used as WWCFs. The WWCF and GnP papers were homogenously mixed in water according to weight percentage ratios (95:5, 90:10, and 85:15), and 1 wt % polyacrylamide solution was added as a binder. Finally, through the filtration process of the WWCF/GnP solution, the respective papers according to their weight percentage ratios were prepared. Subsequently, the WWCF/GnP papers were carbonized at 700, 900, 1100, and 1300 °C. The WWCFs/GnPs paper samples were labeled WWCFs/GnPs-5, WWCFs/GnPs-10, and WWCFs/GnPs-15, according to their GnP content. Figure 1 demonstrates a schematic diagram for preparing WWCF/GnP carbon papers from waste wood.

### 2.3. Analysis of WWCF/GnP Carbon Papers

The crystallinity of the WWCF/GnP carbon papers obtained according to the GnP contents and carbonization temperatures was established by XRD (RIGAKU, D/MAX-2500 instrument, Tokyo, Japan) at operating voltage of 40 kV and current of 30 mA using CuKα radiation. The ordered and disordered structure of the WWCF/GnP carbon papers was obtained by Raman spectra (LabRAM ARAMIS with 514 nm laser, Horiba Jobin Yvon, Tokyo, Japan). The electrical conductivity was determined by measuring the surface conductivity and volume resistivity of the WWCF/GnP carbon papers (LOTESTA-GX MCP-T700, Nittoseiko Analytech, Kanagawa, Japan) in accordance with JIS K 7194 by an ASP type probe using four terminal four pin method, as well as an AC power source (85–264 V, 47–63 Hz, and 40 VA). The EMI SE was performed using a waveguide system with an EMI shielding tester (E5071C ENA Vector Network Analyzer, Keysight, Santa Rosa, CA, USA) scanned in the X-band (8.2–12.4) GHz. Figure 2 shows the EMI shielding analyzer apparatus and the dimensions of carbon paper tips. The test system comprises a Vector Network Analyzer (VNA) of EMI SE analyzers, a coaxial cable, and a coaxial transverse electromagnetic cells as the sample cover. This network analyzer can determine the incident, transmitted, and reflected effectiveness of the WWCF/GnP carbon papers. The tensile strengths of the WWCF/GnP carbon papers using a horizontal length of 50 mm and vertical length of 100 mm were also determined by an Instron 5050 test system (Instron 5050 tester, Norwood, MA, USA). The morphology of the WWCF/GnP carbon papers was observed through SEM (CX-200TA, COXEM, Daejeon, Korea).

## 3. Results and Discussion

### 3.1. X-ray Diffraction Profiles and Raman Analysis

Figure 3 displays the XRD profiles of the WWCF/GnP carbon papers prepared according to the varying GnP contents and carbonization temperatures. All the carbon papers showed broad peaks at around 24–26° and 43°, related to the (002) and (100) planes corresponding to the crystalline structures of the carbonized CFs. The sharp and strong peak at 26° corresponded to the (002) diffraction peak of the GnP. For the WWCF/GnP carbon papers carbonized at 700 °C and 900 °C, the peak intensities of the GnPs were observed, owing to the low growth of the crystalline structure in the carbonized CFs. However, with increasing carbonization temperature, the full width at half-maximum at around 24–26° of 2θ gradually decreased, and the peak at 43° increased. This indicates that the CFs changed into graphite structures with an increase in the carbonization temperature. In addition, two broad and sharp peaks at approximately 24–26° were observed in all the XRD profiles, indicating that the CFs were successfully combined with the GnPs. In addition, the ordered or disordered structure of the WWCF/GnP carbon papers were investigated in the Raman spectra. As shown in Figure 4, the characteristic peaks at 1351 for the D band and 1610 cm^−1^ for G band were observed in all samples. The D-band was a disorder-associated graphite structure and the G-band corresponded to the ordered graphite structure involving the intra-layer vibrations for carbon atoms of sp^2^-bond. Therefore, the peak ratio of the G/D bands (I_G_/I_D_) could be applied to determine graphitic structures. Generally, the I_G_/I_D_ ratio of the WWCF/GnP carbon papers increased as the GnP content and carbonization temperature increased. These results demonstrated that the increased supplementation of GnP and the increased carbonization temperature led to the increase in graphitization.

### 3.2. Electrical Conductivity

Figure 5 shows the electrical conductivity of the WWCF/GnP carbon papers prepared according to the GnP content, carbonization temperature, and number of overlapped carbon paper sheets. As exhibited in Figure 5, the electrical conductivities of the WWCF/GnP carbon papers carbonized at 700 °C were approximately 0 S cm^−1^, regardless of the GnP content and number of overlapped carbon paper sheets. However, as the GnP content and carbonization temperature increased, the electrical conductivities of the WWCF/GnP carbon papers gradually increased. In particular, the number of overlapped carbon paper sheets enhanced the electrical conductivity. The highest electrical conductivity was observed for three overlapped sheets, followed by two sheets and one sheet. For example, WWCF/GnP-15 carbon paper carbonized at 1300 °C overlapped with two sheets had an electrical conductivity value of ≈5.3 S cm^−1^, which further increased to ≈7.09 S cm^−1^ when overlapped with three sheets. These results were obtained because the content with high electrical conductivity GnP (7713 S cm^−1^) and the CF content improved the electrical conductivity, owing to increase in the degree of carbonization with the number of overlapped sheets of WWCF/GnP carbon paper. These electrical conductivity values influence the EMI SE.

### 3.3. Electromagnetic Interference Shielding Effectiveness

When incident EMI waves experience shielding materials, EM waves are reduced due to reflection, absorption, and multiple reflections. Therefore, EMI SE is described as measuring the reduced effect of EM waves by shield materials [16].

As shown in Figure 6, various GnP contents and carbonization temperatures influence the electrical conductivity, which affects the EMI SE of the WWCF/GnP carbon papers. For the WWCF/GnP carbon papers carbonized at 700 °C, most EMI SE values were nearly 0 dB, regardless of the GnP content and number of carbon paper sheets. This was attributed to the low electrical conductivity ≈0 S cm^−1^. In addition, negative EMI SE values were due to the porosity caused by the irregular overlap among fibers of carbon paper, similar to a non-woven fabric [34]. Nevertheless, from carbonization temperature of 900 °C, the EMI SE values increased from approximately 20 to 35 dB at 10.5 GHz with the GnP content increase when a sheet of WWCF/GnP carbon paper was measured. The electrical conductivities were 0.81–1.56 S cm^−1^. Furthermore, the sample thickness can enhance the EMI SE. To study the influence of the thickness of WWCF/GnP carbon paper on EMI SE [26], we measured EMI SE according to the number of sheets. When two sheets of carbon papers were overlapped, the thickness of the WWCF/GnP carbon papers increased, thereby increasing the electrical conductivity (2.9–5.3 S cm^−1^) and the EMI SE ≈42 dB at 10.5 GHz. In addition, the EMI SE values of the WWCF/GnP carbon papers overlapped with three sheets reached approximately 72.5 dB at 10.5 GHz. It is known that 99.9999% of EM radiation can be decreased when the total EMI SE exceeds 60 dB. In the case of WWCF/GnP-15 carbon paper carbonized at 1300 °C, having the electrical conductivity of 7.09 S cm^−1^, the EMI SE values exceeded 60 dB in the range of 8.2–12.4 GHz in the X-band. Due to these high EMI SE values of over 60 dB in the X-band, WWCF/GnPs carbon papers can be used in military and civil applications such as next-generation stealth bodies, gases, and prevention of malfunction in state-of-the-art electronic equipment, or chip-level micro-system, micro inter-system noise suppression/reduction, signal filter parts, etc.

### 3.4. Mechanical Performance of WWCF/GnP Carbon Papers

Figure 7a shows the tensile strengths of the WWCF/GnP carbon papers. Because of the high-temperature treatment, the tensile strength was below 7 MPa and slightly decreased with increasing carbonization temperature, regardless of the GnP content. However, as shown in the optical photographs of Figure 7b–d, the WWCF/GnP-15 carbon paper, having the lowest tensile strength and the highest EMI SE, could be slightly pulled from both the ends, wound, and bent.

### 3.5. Morphology of WWCF/GnP Carbon Papers

The surface and cross-sectional SEM images of the WWCF/GnP carbon papers are shown in Figure 8. Figure 8a illustrates that the GnPs were mechanically well-attached to the WWCF surfaces with polyacrylamide as the binder. These GnPs improved the electrical conductivities and EMI SE values but did not affect the mechanical strengths of the WWCF/GnP carbon papers because they were only attached to the fiber surface, not interconnected among fibers. However, when the GnP content was increased, the GnPs on the surface and in the cross-section of WWCF were distinguished and affixed around the fibers. Thus, the attachment of GnP around the fibers improved the electrical conductivities and EMI SE values.

## 4. Conclusions

In conclusion, we isolated WWCFs, without completely removing contaminants, for recycling and converted them into highly valuable EMI shielding materials. According to the WWCF and GnP content, the WWCF/GnP papers were manufactured, and the subsequent WWCF/GnP carbon papers were prepared via carbonization for application as EMI shielding materials. The GnP contents and carbonization temperatures influenced the electrical conductivity and EMI SE. Furthermore, the increase in the GnP content and carbonization temperature increased the electrical conductivity and the EMI shielding in the high-frequency X-band. Moreover, the EMI SE value of the WWCF/GnP-15 carbon paper carbonized at 1300 °C was 43 dB of at 10.5 GHz in the case of one sheet. When three sheets were overlapped, the EMI SE values of the WWCF/GnP carbon papers reached approximately 72.5 dB at 10.5 GHz.

## Figures and Tables

**Figure 1 nanomaterials-11-02878-f001:**
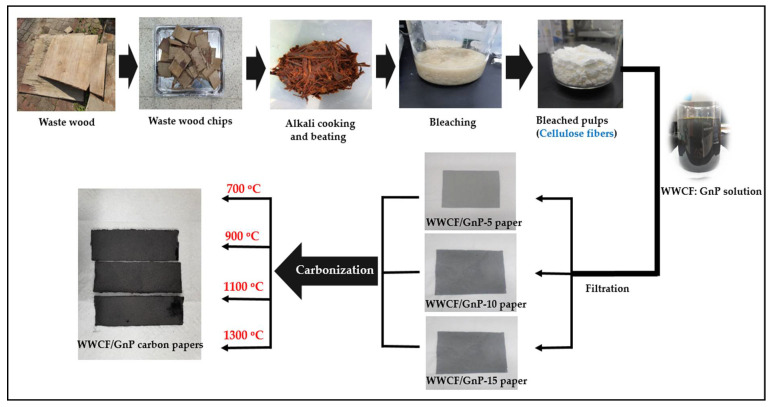
Schematic diagram to prepare waste wood cellulose fiber/graphene nanoplatelet (WWCF/GnP) carbon papers.

**Figure 2 nanomaterials-11-02878-f002:**
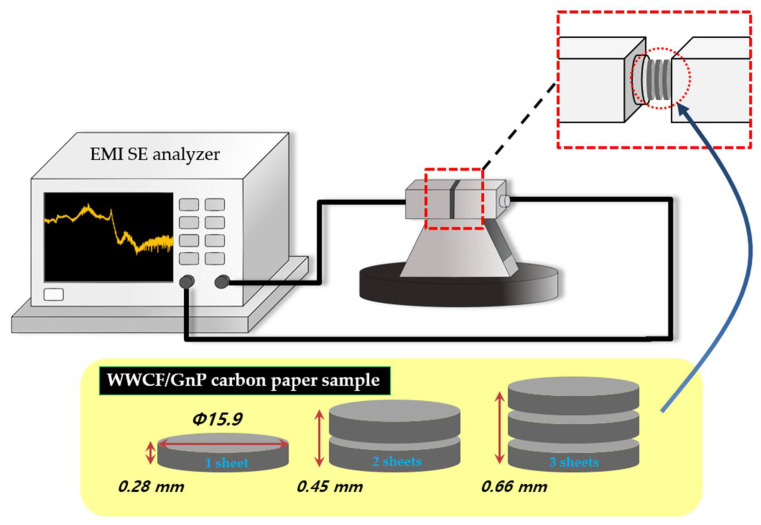
Apparatus of electromagnetic interference (EMI) shielding (EMI SE, EMI shielding effectiveness; WWCF/GnP, waste wood cellulose fiber/graphene nanoplatelet).

**Figure 3 nanomaterials-11-02878-f003:**
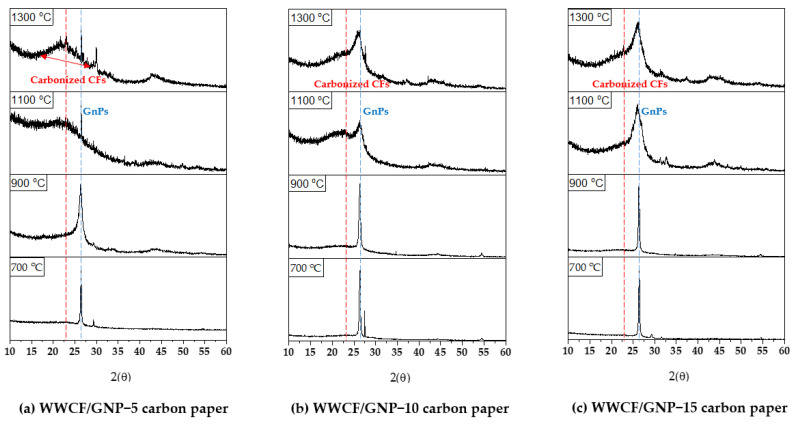
X-ray diffraction (XRD) patterns of waste wood cellulose fiber/graphene nanoplatelet (WWCF/GnP) carbon papers.

**Figure 4 nanomaterials-11-02878-f004:**
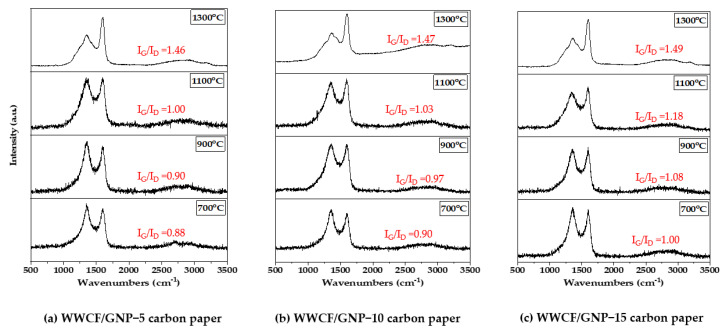
Raman spectra of waste wood cellulose fiber/graphene nanoplatelet (WWCF/GnP) carbon papers.

**Figure 5 nanomaterials-11-02878-f005:**
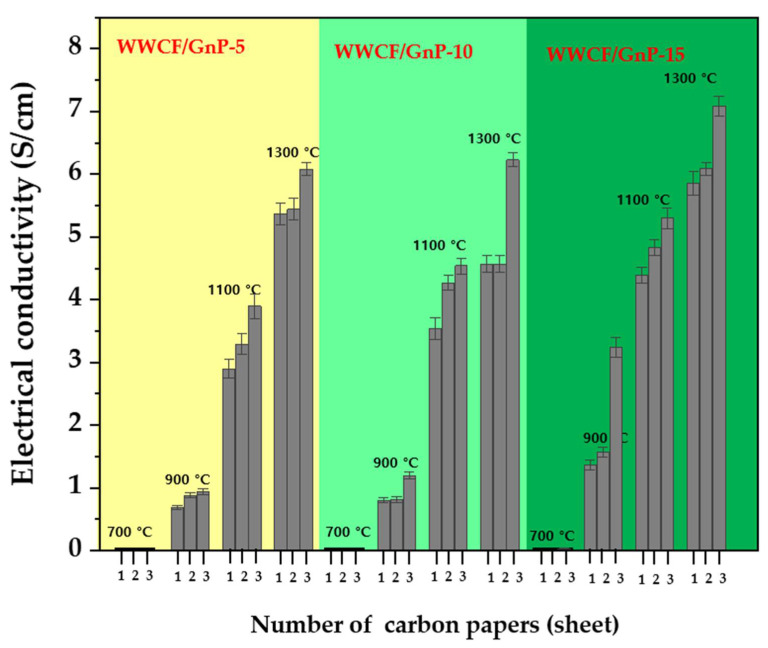
The electrical conductivity comparison of waste wood cellulose fiber/graphene nanoplatelet (WWCF/GnP) carbon papers prepared according to the GnP content, carbonization temperature, and number of overlapped carbon paper sheets.

**Figure 6 nanomaterials-11-02878-f006:**
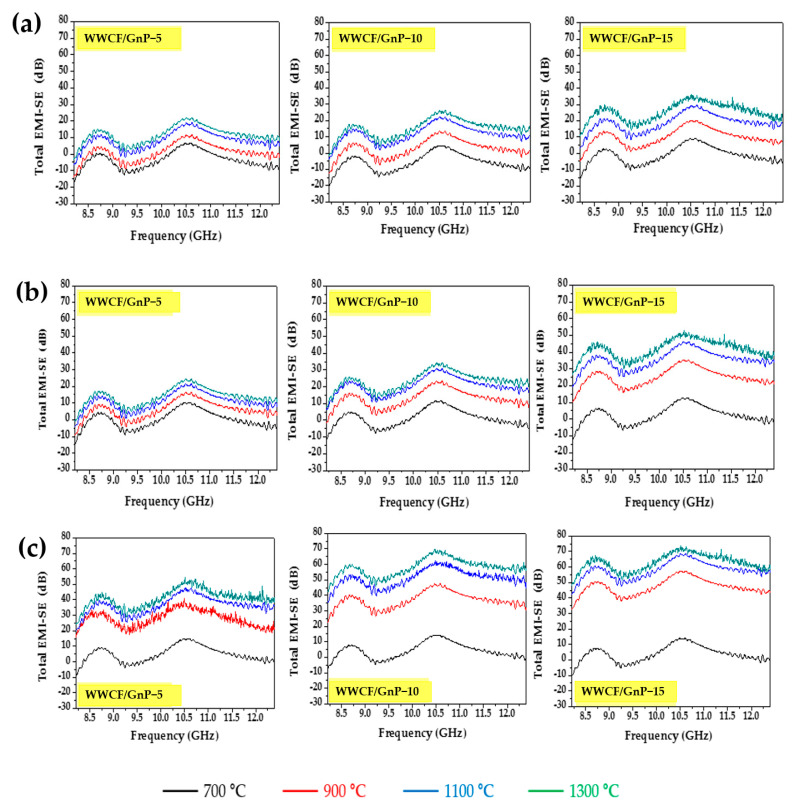
Electromagnetic interference shielding effectiveness (EMI SE) of waste wood cellulose fiber/graphene nanoplatelet (WWCFs/GnPs) carbon papers prepared according to the GnP contents and carbonization temperatures in the case of one sheet (**a**), two sheets (**b**), and three sheets (**c**).

**Figure 7 nanomaterials-11-02878-f007:**
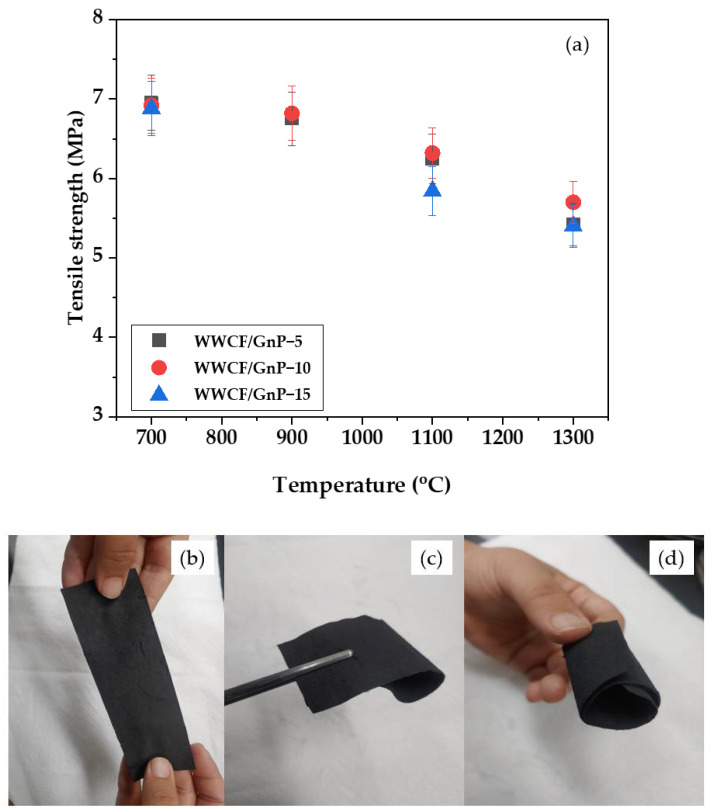
Tensile strengths of waste wood cellulose fiber/graphene nanoplatelet (WWCF/GnP) carbon papers prepared according to the varying GnP contents and carbonization temperatures (**a**), and photographs of the pulled shape to both sides (**b**), the wound shape (**c**), and the bent shape (**d**).

**Figure 8 nanomaterials-11-02878-f008:**
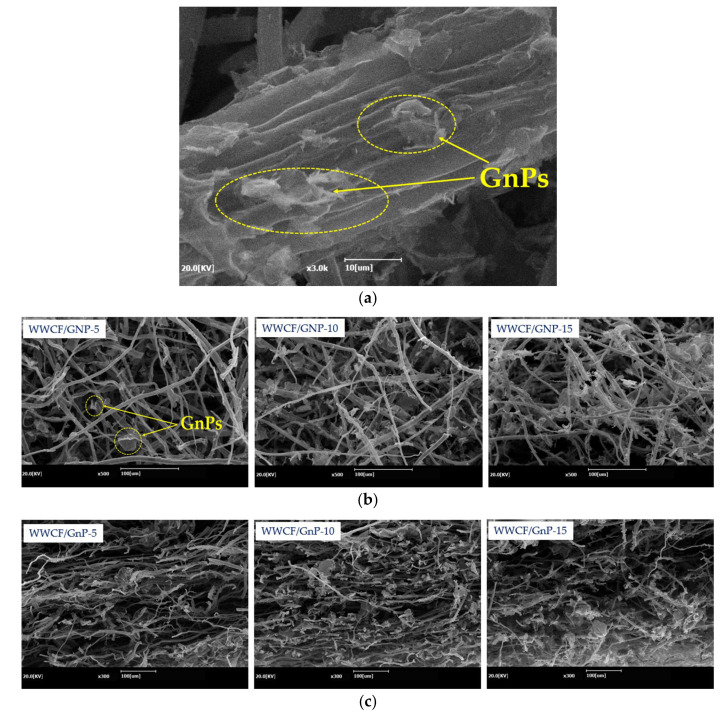
Scanning electron microscopy (SEM) images for (**a**) magnification: 3000×, (**b**) the surface, and (**c**) cross-section of waste wood cellulose fiber/graphene nanoplatelet (WWCF/GnP) carbon papers.

## Data Availability

The data are available by corresponding author.

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
