# Peer review of "Fabrication and Characterization of Waste Wood Cellulose Fiber/Graphene Nanoplatelet Carbon Papers for Application as Electromagnetic Interference Shielding Materials"

_nanomaterials, 2021, doi:10.3390/nano11112878_

Round 1

Reviewer 1 Report

The topic of using natural materials for EMI shielding is important and in line with research priorities in the field.    There are places where editing of English is needed and I marked those sentences in yellow hilite on the document.

While the results are clearly presented in most cases, I have some concerns about the lack of explanations/interpretation for the results.  The paper would be of much greater value if the authors did more than just present the data.  Specific suggestions and questions follow.

  • Line 155 - what is "low volume resistivity".  The important parameters impacting EMI are surface and thru thickness conductivity.  This needs clarification.
  • Line 187 - What is meant by CF being "combined" with GnP?  Is this a chemical bond, mechanical connection or agglomeration?
  • Fig 5 and associated writeup - Need to clarify if this is thru thickness conductivity.  If yes, the data represents a combination of inidvidual sheet resistance and interface resistance between sheets.  However, there is high variability on adding extra sheets.  The authors should provide an analysis of the conductance of individual sheets and then measure what happens when they are stacked.  A simple model analysis should be added to the paper.  Also provide an explanation for the variability. 
  • Figure 6 - For some samples, the data is showing a negative shielding efficiency.  How is this possible.  The authors need to provide more details on how they are doing their measurement and interpreting the data.  VNA experiments typically require specimens that have parallel and flat sample sides.  How are they making their samples for the VNA.
  • Also important for the authors to expand quantitatively on the relationship between conductance values and EMI results.

By addressing these issues, the authors will be providing the reader with explanations and interpretations of the data, vs just presenting the results.

Author Response

Dear reviewer

We are thankful to you for your constructive comments on this manuscript.

Accordingly, the following revisions were made in the manuscript.

The topic of using natural materials for EMI shielding is important and in line with research priorities in the field.    There are places where editing of English is needed and I marked those sentences in yellow hilite on the document.: I have revised the lines marked in yellow.

  • Line 155 - what is "low volume resistivity".  The important parameters impacting EMI are surface and thru thickness conductivity.  This needs clarification.

 → I have revised “low volume resistivity” to “volume resistivity” and added the description “in accordance with JIS K 7194 by an ASP type probe using four terminal four pin method and the power source was (85-264 V 47-63 Hz and 40 VA).”

  • Line 187 - What is meant by CF being "combined" with GnP?  Is this a chemical bond, mechanical connection or agglomeration?

→ WCF and GnP combine via mechanical connections. As per your comment, I have added the SEM image and the following explanatory sentence “Figure 8 (a) illustrates that GnPs were mechanically well-attached to the WWCF surfaces with polyacrylamide as the binder.”

  • Fig 5 and associated writeup - Need to clarify if this is thru thickness conductivity.  If yes, the data represents a combination of inidvidual sheet resistance and interface resistance between sheets.  However, there is high variability on adding extra sheets.  The authors should provide an analysis of the conductance of individual sheets

→ As shown in Figure 5, the mean electrical conductivity value for a carbon papers is represented by a bar graph and the deviation is represented by error bars. The same applies to the electrical conductivities of two and three sheets.

and then measure what happens when they are stacked.  A simple model analysis should be added to the paper.  Also provide an explanation for the variability. 

→ As per your comment, I have added the sentence  “ These results were obtained because the GnP content with high electrical conductivity (7713 S/cm) and the CF content improved the electrical conductivity owing to increase in the degree of carbonization with the number of overlapped sheets of WWCF/GnP carbon paper.”

  • Figure 6 - For some samples, the data is showing a negative shielding efficiency.  How is this possible. The authors need to provide more details on how they are doing their measurement and interpreting the data.

→ I have revised as per your comments.

After: In addition, the negative EMI SE values were due to the porosity caused by the irregular overlap among the fibers of carbon paper similar to a non-woven fabric (Figure 8) [34].

VNA experiments typically require specimens that have parallel and flat sample sides.  How are they making their samples for the VNA.

→ We prepared the WWCF/GnP carbon paper sample as shown in Figure 2.

  • Also important for the authors to expand quantitatively on the relationship between conductance values and EMI results.

→ I have quantitatively explained the relationship between electrical conductivity and EMI results, as your per comments.

Thank you again for your valuable comments and insightful suggestions.

Best regards.

Dr. Hye Kyoung Shin

Reviewer 2 Report

Authors reported the synthesis of materials of waste wood. In this study, waste wood cellulose fiber/graphene nanoplatelet papers by using carbonization variation, and investigated their EMI shielding property. Materials realization and experimental characterization are properly given; the results are clearly presented. However, some issues should be addressed.

1, Authors should figure out the significance and real-life application of the present paper in Abstract section.

2, In Experimental section, the characterization of EMI property should be more elaborated, such as the sample size, test method, and test environment, so that readers can judge the accuracy of the present data. In XRD pattern, the character peaks of cellulose fiber and graphene should be pointed out and index in the Figure 3.

3, Is the material (waste wood cellulose fiber/ graphene nanoplatelet papers) obtained from physical mixing or in-situ growth? It’s not clearly mentioned in the manuscript. While, I think that the effect of the polarization to improve the electromagnetic wave absorption and shielding will be neglected if the material just be obtained by physical mixing. Authors should explain the interaction between waste wood cellulose fiber and graphene nanoplatelet, which would affect both the shield capability and mechanical property.

4, Some key and important research results about graphene and its electromagnetic property should be mentioned and cited so that we can provide a solid background and progress to the readers, such as Journal of Materials Chemistry C, 2016, 4, 9738; ACS Applied Materials & Interfaces, 2017, 9, 16404; Composites Part A, 2018, 115, 371.

5, Since the layers of composite papers have great influence on the EMI shielding and conductive property, there must have been some reason why authors chose different paper sheets to compare their performance. Authors should give the reason.

6, The mechanical property of samples is too low to satisfy the real-life application. How to achieve the strength requirements in next step, please give more details.

Author Response

Dear reviewer

 We are thankful to you for your constructive comments on this manuscript.

Accordingly, the following revisions were made in the manuscript.

1, Authors should figure out the significance and real-life application of the present paper in Abstract section.

 → I have added a sentence in the Abstract as per your comment.

After: the WWCF/GnP papers were varyingly carbonized for their application as electromagnetic interference (EMI) shielding materials in military and civil applications such as state-of-the-art electronic equipment malfunction prevention, chip-level microsystem, and micro intersystem noise suppression/reduction.

2, In Experimental section, the characterization of EMI property should be more elaborated, such as the sample size, test method, and test environment, so that readers can judge the accuracy of the present data.

 → I have added the following sentence “Figure 2 shows the EMI shielding analyzer apparatus and the dimensions of carbon paper tips. The test system comprises a Vector Network Analyzer (VNA) of EMI SE analyzers, a coaxial cable, and a coaxial transverse electromagnetic cells as the sample cover.”

In XRD pattern, the character peaks of cellulose fiber and graphene should be pointed out and index in the Figure 3.

→ I have revised the figure as per your comments.

After:

3, Is the material (waste wood cellulose fiber/ graphene nanoplatelet papers) obtained from physical mixing or in-situ growth? It’s not clearly mentioned in the manuscript. While, I think that the effect of the polarization to improve the electromagnetic wave absorption and shielding will be neglected if the material just be obtained by physical mixing. Authors should explain the interaction between waste wood cellulose fiber and graphene nanoplatelet, which would affect both the shield capability and mechanical property.

 → I have added SEM image of Figure 8(a) and the sentence.

After: Figure 8 (a) illustrates that the GnPs were mechanically well-attached to the WWCF surfaces with polyacrylamide as the binder. These GnPs improved the electrical conductivities and EMI SE values but didn’t affect the mechanical strengths of the WWCF/GnP carbon papers because they were only attached to the fiber surface, not interconnected among fibers.

4, Some key and important research results about graphene and its electromagnetic property should be mentioned and cited so that we can provide a solid background and progress to the readers, such as Journal of Materials Chemistry C, 2016, 4, 9738; ACS Applied Materials & Interfaces, 2017, 9, 16404; Composites Part A, 2018, 115, 371.

→ Thank you for recommending a pertinent references. The first paper was cited [Ref. 35] and the second and third papers were added as References [19 and 20, respectively].

5, Since the layers of composite papers have great influence on the EMI shielding and conductive property, there must have been some reason why authors chose different paper sheets to compare their performance. Authors should give the reason.

→ I have given the reason as per your comments

After: Furthermore, the sample thickness can enhance the EMI SE. To study the influence of the thickness of WWCF/GnP carbon paper on EMI SE [26], EMI SE was measured according to the number of sheets. 

6, The mechanical property of samples is too low to satisfy the real-life application. How to achieve the strength requirements in next step, please give more details.

→ Although the mechanical property, i.e., tensile strength of the WWCF/GnP carbon papers is very low, the samples didn’t easily break, which is evident from Figure 7(b), (c), and (d). Furthermore, with respect to enhancing the mechanical property, we are researching on how to change the binder or develop it into a composite.

Thank you again for your valuable comments and insightful suggestions.

Best regards.

Dr. Hye Kyoung Shin

Round 2

Reviewer 1 Report

The authors have addressed the concerns raised in the earlier review